# Microfluidic Synthesis of Vinblastine-Loaded Multifunctional Particles for Magnetically Responsive Controlled Drug Release

**DOI:** 10.3390/pharmaceutics11050212

**Published:** 2019-05-03

**Authors:** Keng-Shiang Huang, Chih-Hui Yang, Ya-Chin Wang, Wei-Ting Wang, Yen-Yi Lu

**Affiliations:** 1The School of Chinese Medicine for Post-Baccalaureate, I-Shou University, Kaohsiung 82445, Taiwan; Yachinwang1219@gmail.com (Y.-C.W.); sheep8068@hotmail.com (W.-T.W.); a2828301@gmail.com (Y.-Y.L.); 2Department of Biological Science and Technology, I-Shou University, Kaohsiung 82445, Taiwan; 3Taiwan Instrument Research Institute, National Applied Research Laboratories, Hsinchu 30076, Taiwan

**Keywords:** microfluidics, superparamagnetic, chitosan, vinblastine, drug delivery, controlled release

## Abstract

Vinblastine (VBL) is a major chemotherapeutic drug; however, in some cases, it may cause severe side effects in patients with cancer. Designing a novel VBL pharmaceutical formulation is a crucial and emerging concern among researchers for reducing the use of VBL. This study developed a stimuli-responsive controlled VBL drug release system from magnetically sensitive chitosan capsules. A magnetically responsive controlled drug release system was designed by embedding superparamagnetic iron oxide (SPIO) nanoparticles (NPs) in a chitosan matrix and an external magnet. In addition, droplet microfluidics, which is a novel technique for producing polymer spheres, was used for manufacturing monodispersed chitosan microparticles. The prepared VBL and SPIO NPs-loaded chitosan microparticles were characterized and analyzed using Fourier transform infrared spectroscopy, transmission electron microscopy, scanning electron microscopy, a superconducting quantum interference device, and a biocompatibility test. The drug encapsulation efficiency was 67%–69%. The in vitro drug release test indicated that the VBL could be 100% released from chitosan composite particles in 80–130 min under magnetic stimulation. The pulsatile magnetically triggered tests showed individual and distinctive controlled release patterns. Thus, the timing and dose of VBL release was controllable by an external magnet. The results presume that using a magnetically responsive controlled drug release system offers a valuable opportunity for VBL drug delivery, where the delivery system is an active participant, rather than a passive vehicle, in the optimization of cancer treatment. The proposed actively targeted magnetic drug delivery system offers many advantages over conventional drug delivery systems by improving the precision and timing of drug release, easy operation, and higher compliance for pharmaceutical applications.

## 1. Introduction

At present, vinca alkaloids are crucial for cancer therapy. Vinblastine (VBL), a vinca alkaloid, has been widely used in chemotherapy [1,2]. VBL acts on intracellular tubulin to inhibit cell division by blocking mitosis, purine and RNA synthesis to rapidly destroy the dividing cells [3]. VBL is used to treat several types of cancer, such as lymphocytic lymphoma, histiocytic lymphoma, testicular cancer, breast cancer, Kaposi’s sarcoma, and Hodgkin’s disease [4,5]. Side effects of VBL are the toxicity to white blood cells, constipation dyspnea, chest pain, vomiting, and antidiuretic hormone secretion. The risk of side effects has limited the development of VBL in clinical applications [1,3]. Therefore, several types of drug delivery system have been developed for reducing these side effects [6,7]. For example, i) thymidine-conjugated VBL, a bifunctional molecule, forms a microtubule-binding agent and disease-associated kinase; this is the first report on kinase-mediated trapping of cancer therapeutics [4,5,8]. ii) Folic acid peptide-bound desacetylvinblastine hydrazide and carbohydrate-based VBL–folate could specifically target tumors that overexpress the folate receptor; however, concerns exist about the desacetyl vinblastine hydrazide occurred high frequency to induce constipation in patients [9,10]. iii) The VBL sulfate-loaded folate-conjugated bovine serum albumin nanoparticles (NPs) as a new targeted antitumor drug eliminates the side effects of VBL sulfate [11]. iv) VBL sulfate encapsulated in poly(lactide-*co*-glycolide) microspheres was stable for more than a month [12]. v) Liposome-encapsulated VBL microspheres are present in different forms, such as cationic liposomes [13], anionic liposomes [14], and multilamellar vesicle [15]. VBL was encapsulated in the aqueous core of liposomes, and in vivo results revealed that the targeting liposomes could have a prolonged circulation time in the blood circulation system, effectively terminate cancer cells, and reduce the toxicity of VBL to normal cells [16]. vi) VBL was incorporated with magnetic NPs into cationic liposomes to achieve the antitumor effect for reducing tumor nodules and targeting the tumor vasculature [17].

Currently, a magnetically responsive system is a novel and emerging nanotechnology strategy for developing drug-controlled release systems and nanomedicines [18,19]. Superparamagnetic iron oxide (SPIO) NPs (e.g., Fe_3_O_4_ NPs) is typically the most commonly used materials in magnetically responsive systems [20,21,22]. SPIO NPs have high chemical stability under physiological conditions and biocompatibility with low toxicity, and they have been approved by the US Food and Drug Administration. The advantages of a magnetically responsive system are i) the high drug release rate under a magnetic field (using the collapse or volume transition of the drug spheres to promote drug release); ii) controllable timing and the drug release amount; iii) ability to provide magnetic guidance under a permanent magnetic field; and iv) possible application in magnetic resonance imaging and thermo chemotherapy [23,24,25]. We presume that the advantages and distinguishing characteristics of magnetic guidance and magnetically controlled drug release could overcome the limitations raised by the aforementioned conventional drug delivery systems, such as the high polymer cost, low drug entrapment efficiency, low releasing rate, or broad size distribution of drug carriers. The targeted delivery of pharmacologically active molecules could be magnetically guided to a carcinoma area in the body by using magnetic iron oxide NPs under magnetic field gradients [26]. In addition, the magnetically responsive controlled drug release system could provide compressive and tensile stresses or a pulsatile deformation to polymers under a magnetic field [27]. For example, Kost et al. conducted pulsatile release from polymer composites by using external magnetic fields and externally controlled insulin release from a magnetic ethylene–vinyl acetate copolymer [28]. Magnetically guidable NPs have also been used to deliver nucleic acid, enzymes, genes, DNA vaccine, siRNA, or anticancer drugs by using an external magnet for reducing multidrug resistance, avoiding drug degradation during delivery, or achieving effective silencing effects [29]. 

Recently, we have successfully established a platform for manufacturing various uniform drug carriers through droplet microfluidics [30,31]. The ultra-uniform microparticles could enable a more efficient administration of a controlled drug release system. The size of the polymer particles could be tuned by adjusting the flow rates of the water and oil phases in the range of tens to hundreds of micrometers in diameter [32]. In addition, functional drug carriers could be obtained by combining NPs with polymers. Herein, we designed a magnetically responsive functional drug carrier by loading SPIO NPs in a chitosan polymer [33]. The magnetically responsive drug formulation was prepared with a uniform particle size by using a microfluidic chip, and it was further applied in the controlled VBL drug release study. An external magnet was used to exploit the magnetically guidable property of chitosan composites and conduct pulsatile VBL drug release from chitosan composites.

The magnetically guidable property was designed for concentrating drug carriers in a specific region. When the aggregated drug carriers release the drug, the local concentration of VBL would increase and subsequently benefit anticancer treatment. The proposed magnetically responsive controlled drug release system was designed for reducing the side effects and cellular toxicity of VBL. The proposed magnetically responsive controlled VBL drug release system provides an alternative method for cancer therapy.

## 2. Materials and Methods 

### 2.1. Materials

Chitosan (molecular weight (MW), 150,000 and 400,000), VBL sulfate, glycine, and acetic acid (CH_3_COOH) were purchased from Sigma–Aldrich Co. (St. Louis, MO, USA). Iron (II) chloride tetrahydrate (FeCl_2_N_4_H_2_O, 99% purity) was purchased from Backer Inc. (Phillipsburg, NJ, USA). Iron (III) chloride hexahydrate (FeCl_3_N_6_H_2_O, 97% purity) was purchased from Alfa Aesar (Ward Hill, MA, USA). Sodium hydroxide (NaOH) was purchased from Macron Fine ChemicalsTM (Center Valley, PA, USA). Distilled water was filtered using a 0.22 nm syringe filter (Millipore Inc., Clifton, NJ, USA) before use in the preparation process. All reagents are of the highest grade and commercially available.

### 2.2. Design and Fabrication of a Microfluidic Chip 

The microfluidic chip was designed by AutoCAD 2007 (Autodesk, San Rafael, CA, USA, USA); it was laid out on a conventional polymethyl methacrylate substrate (length, 86 mm; width, 43 mm; and depth, 2 mm) by using a computer-controlled CO_2_ laser machine (LaserPro Venus, GCC, Taiwan) [34,35]. The microfluidic channel chip shown in Figure 1A comprised three layers: the cover layer (containing 3 inlet ports and 20 screw holes), main layer (containing the cross-junction microchannels and screw holes), and bottom layer (containing one outlet port and screw holes). The three layers were integrated using screws (pitch, 0.5 mm and diameter, 4 mm) to produce a microfluidic chip (Figure 1B). This design of microfluidic chip is similar to the chips reported in our previous studies [36]. This microfluidic chip was highlighted in the facile fabrication process and disassembly module of screws. Each chip could be reused and cleaned after the tight screws were loosened.

### 2.3. Synthesis of the Chitosan Particles

A chitosan solution, which contained 0.1 g of chitosan dissolved in 2 mL of 1% (*v*/*v*) CH_3_COOH solution (99.5 wt.%), was loaded into a syringe. Syringe pumps were subsequently used to pump both the chitosan solution and oil onto microchannels of the microfluidic chip. After chitosan droplets were generated on a microfluidic chip, then transported, and dropped into a 20% NaOH solution. After 10 min, chitosan particles were observed. Spheres were collected through centrifugation and were washed twice with 30 mL of distilled water to remove alkali (Figure 2).

### 2.4. Synthesis of SPIO NPs–Chitosan Composite Particles

A ferrous solution containing 4.776 g of FeCl_2_·4H_2_O was dissolved in 12 mL of a 2 N HCl solution; a ferric solution containing 3.24 g of FeCl_3_·6H_2_O was dissolved in 12 mL of a 2 N HCl solution. Ferrous and ferric solutions were prepared by dissolving FeCl_2_ and FeCl_3_, respectively, in 1:4 molar ratios and mixed by constant stirring for 30 min. Furthermore, 5 mL of ferrous and ferric solutions and 5 mL of a glycine solution (1 g/mL, dissolved in distilled water) were mixed. A 20% NaOH solution slowly dropped in ferrous and ferric solutions which contained glycine until the color of the solution turned black to obtain an SPIO NPs solution (e.g., Fe_3_O_4_ NPs solution).

Three milliliters of the chitosan solution and 1 mL of the SPIO NPs solution constituted the SPIO NPs–chitosan solution, which was loaded into a single output syringe as the disperse phase. The continuous phase comprised sunflower seed oil and was loaded into a double output syringe. Syringe pumps (KDS Model 220 Series; Kd Scientific, Greater Boston Area, East Coast, New England) were used to simultaneously inject both sample flow (disperse phase) and sheath flow (continuous phase) through Teflon tubes into the microfluidic chip. A spontaneous self-assembly water-in-oil emulsion was obtained at the cross-junction by using two continuous oil streams to shear the SPIO NPs–chitosan solution. Emulsion droplets were dripped into a NaOH solution for the solidification. After 10 min, SPIO NPs–chitosan composite particles were observed. The black particles were subsequently collected through centrifugation and washed several times with 30 mL of distilled water to remove any alkali.

### 2.5. Preparation of VBL-Loaded SPIO NPs–Chitosan Composite Particles

Four mL of a VBL solution was prepared by dissolving 80 mg of VBL powder in 4 mL of distilled water, then mixed it with 4 mL of the SPIO NPs–chitosan solution, and used the mixture solution as the sample flow (disperse phase). The microfluidic synthesis procedure of VBL-encapsulated SPIO NPs-loaded chitosan particles was the same to that described in the previous section. The fabricated particles were subsequently collected through centrifugation and washed twice with 30 mL of double distilled water to remove any residual VBL and alkali.

### 2.6. Cytotoxicity Test

The viability tests were measured using the 3–(4,5–Dimethylthiazol–2–yl)–2,5–diphenyltetrazolium bromide (MTT) assay with MCF-7 and NIH 3T3 cells, respectively. The cells with a 1 × 10^4^/well density were seeded into a 96-well plate containing 100 µL of a culture medium (RPMI 1640, DMEM) and subsequently allowed to adhere for 24 h. They were then treated with various concentrations of the VBL-loaded SPIO NPs–chitosan composite particles. After 24 h of exposure, a 200-µL MTT solution (1 mg/mL) was added for conducting a 4-h reaction with the cells. After removing the medium and MTT solution, 100 µL of dimethyl sulfoxide was added to each well, and then the assay plate was read at 595 nm by using a microplate reader (Multiskan Ascent, Thermo Electron, Waltham, MA, USA). The absorbance of the control cells was considered to be 100%.

### 2.7. Application in Magnetically Responsive Drug Release

In vitro drug release data revealed the VBL release profile from the VBL-loaded SPIO NPs–chitosan composite particles with or without a pulsatile magnetic field. These composite particles were immersed in a water bath at 37 °C. A pulsatile magnetic field could be generated by intermittently attaching (distance < 0.1 cm) or detaching (distance > 2 meters) the magnets (4 cm × 2 cm × 2.5 cm, strength 4455 ± 255 gauss) to the vessel. Each magnetic field pulsatile sequence lasted for 10 min at the 20th, 40th, 60th, 80th, 100th, and 120th min, respectively.

## 3. Results

### 3.1. Morphology

The microfluidic chip can effectively control the particle diameter and produce uniform particles with a coefficient of variation of a diameter distribution of less than 6.2%, indicating the manufactured prepared particles were monodispersed [37]. The chitosan particles (chitosan, 400,000 MW) were clearly visible through optical microscopy (Figure 3A–C), and their size was approximately 300 µm. Figure 3D–F illustrates the optical images of SPIO NPs–chitosan composite particles. We presumed the formation of SPIO NPs inside the chitosan particles because the particles appeared black. On the other hand, chitosan particles appeared transparent because SPIO NPs were absent [38,39]. Figure 4 represents scanning electron microscopy images of the VBL-loaded chitosan composite particles with or without SPIO NPs. All VBL-loaded SPIO NPs–chitosan composite particles revealed desirable roundness and surface smoothness (Figure 4A–D). Furthermore, chitosan particles contained SPIO NPs exhibited favorable rigidity and higher mechanical strength. Figures show in 4B and 4D indicating that the surface of SPIO NPs–chitosan composite particles is filled with compact granules. 

The effects of experimental parameters on the morphology of particles by using a microfluidic chip are shown in Table 1. The flow rates of continuous and dispersed phase considerably affected the size of the synthesized beads. The diameter of the composite emulsions was in the ranged from 508 to 560.6 µm. The relative standard deviation (R.S.D.) was less than 10% (from 0.8% to 3.8%), indicating each emulsion was uniform in size. After solidification, the average size of composite particles was ranged from 359.4 to 423.1 µm. The shrinking rate from the emulsion stage to the particle stage was 25.6% to 30.4%. In addition, the R.S.D. was extremely low (2.8% to 66.5%) in the particle stage, meeting the typical criterion for monodispersity [40]. The microsphere size can be flexibly controlled by varying the flow rate of the discontinuous and dispersed phases. These results are consistent with our previous study results [41]. Therefore, we can successfully fabricate uniform VBL-loaded SPIO NPs–chitosan composite particles with different sizes by changing the flow rates. 

Generally, the uniformity and particle size of drug carriers influence the drug release rate, release pattern, and release kinetics. Uniform particles ultimately determine the drug-packaging dosage and release efficiency as well as promote the drug delivery system quality [35]. The droplet microfluidic is a novel and emerging tool for manufacturing emulsions with precisely controlled and monodisperse size distributions [42]. Based on the results of Table 1, we could prepare uniform morphology drug carriers form chitosan polymer [36]. The monodispersed chitosan particles could provide better administration of controlled drug release than conventional dispersed drug formulations. The chitosan particles were functionalized by embedding SPIO NPs in a chitosan matrix. The prepared chitosan composite particles could be magnetically guided by a magnet. Therefore, we presume that these composite particles could be applied as a magnetically responsive controlled drug release system. The VBL would be chosen as a model drug for the stimuli-responsive controlled VBL drug release system from magnetically sensitive chitosan capsules.

### 3.2. Characterization

Figure 5 indicates the Fourier transform infrared spectroscopy (FTIR) of chitosan particles, SPIO NPs–chitosan composite particles (MW, 150,000), SPIO NPs–chitosan composite particles (MW, 40,000), and VBL-loaded SPIO NPs–chitosan composite particles (MW, 150,000), respectively. In the spectrum of chitosan (curve A), the characteristic absorption peaks appeared at 3380 cm^−1^, 2874 cm^−1^ and 1653 cm^−1^ indicated the vibrations of free amine groups, stretching of –CH_3_ group and C=O stretching, respectively. These characteristic peaks were found in SPIO NPs–chitosan composite particles (chitosan MW, 150,000; curve B), SPIO NPs–chitosan composite particles (chitosan MW, 40,000; curve C), and VBL-loaded SPIO NPs–chitosan composite particles (curve D), indicating the presence of chitosan. 

Comparing the chitosan particles and SPIO NPs-embedded chitosan composite particles (curve A versus to curves B and C), results show that an absorption peak appears at about 550–650 cm^−1^ which attributed to the Fe–O group, indicating that iron oxide particles were embedded in the chitosan particles. In addition, the FTIR spectra of SPIO NPs-embedded chitosan composite particles with two kinds MW of chitosan polymers were very similar (please see curve B and curve C in Figure 5), indicating various chitosan polymers could be used as a drug carrier. 

Figure 6A shows transmission electron microscopy (TEM) image of the prepared composite particles, result shows that the prepared iron oxide particles are in nanosize. Because of the high electron density of the iron oxide NPs [43], the dark areas indicated that the aggregated particles with a diameter of 7 nm. In Figure 6B, the superconducting quantum interference device (SQUID) was used to measure iron oxide NPs, which indicated that iron oxide NPs are superparamagnetic components and the saturation magnetization (Ms) value of SPIO NPs–chitosan composite particles was 20 emu/g. Therefore, we identified the synthesized particles as SPIO NPs–chitosan composite particles. Because the prepared composite particles with high Ms value, this superparamagnetic property renders the composite particles more susceptible to the external magnetic field by a magnet.

When compare the FTIR spectra among the curves B, C, and D in Figure 5, two distinguishing absorption peaks around 2930 cm^−1^ (overlapping C–H stretching vibrations of methyl, methylene and –CH) and 1689 cm^−1^ were observed (stretching vibration peak of C=O group), indicating the loading of VBL drug inside the composite particles [44]. Based on the high-performance liquid chromatography data, the VBL drug encapsulation efficiency was 67%–69%. 

### 3.3. Biocompatibility Test and In Vitro Release Study

The biocompatibility of the composite particles was evaluated using the MTT assay by incubating 3T3 and MCF-7 cell lines with the prepared SPIO NPs–chitosan composite particles (0–1000 µg/mL) for 24 h. The results revealed high viability (> 80%) even under a high concentration of the SPIO NPs–chitosan composite particles (1000 µg/mL). Figure 7 show that the SPIO NPs-embedded chitosan capsules are potential biocompatible materials and has satisfactory stability to deliver VBL [45,46], revealing that the prepared particles were potential biocompatible materials and have potential to be applied in pharmaceutical fields. 

### 3.4. Application in Controlled Drug Release

Generally, chitosan with different MW may affect the drug release profile 43. Two kinds MW of chitosan polymers were used to test the in vitro controlled drug release. In addition, the in vitro drug release profile of the VBL-loaded SPIO NPs–chitosan composite particles were controlled using a magnet for 10 min at the 20th, 40th, 60th, 80th, 100th, and 120th min, respectively (as shown in Figure 8). The magnetic field was created using a magnet, which was placed apart on the side of the vessel. A pulsatile magnetic field could be generated by placing the magnet close to the vessel or moving them away 40. Results show that the cumulative drug release rate of low MW (150,000) chitosan polymer is faster than the high MW (400,000) one. For example, the accumulative drug release rate of the VBL-loaded SPIO NPs–chitosan (MW, 150,000) particles reached 100% at the approximately 90th min under magnetic stimulation application; the accumulative drug release rate of the VBL-loaded SPIO NPs–chitosan (MW, 400,000) particles reached 100% at the approximately 130th min under magnetic stimulation application. In these cases, a 40 min significant difference was observed between two kinds MW of chitosan polymers (please see black triangle lines in Figure 8). The same tendency was found in without the magnetic stimulation cases (please see white circle lines or black circle lines in Figure 8). 

In the magnetic stimulation tests, results show that the cumulative drug release rate of magnetic stimulation is faster than without the magnetic stimulation one. For example, when four pulses of magnetic stimulation to the VBL-loaded SPIO NPs–chitosan (MW, 150,000) particles, the accumulative drug release rate reached 100% at the approximately 90th min. In the same time (90th min) and without the magnetic stimulation case, it only reached an accumulative drug release rate of approximately 45%. The results show that a significant difference (55% of the cumulative drug release rate) was observed between with and without the magnetic stimulations (please see black triangle lines versus white circle lines at 90th min in Figure 8A). The same tendency was found in high MW (400,000) chitosan polymer case (please see black triangle lines versus white circle lines at 130th min in Figure 8B). When six pulses of magnetic stimulation to the VBL-loaded SPIO NPs–chitosan (MW, 400,000) particles, the accumulative drug release rate reached 100% at the approximately 130th min. In the same time (130th min) and without the magnetic stimulation case, the accumulative drug release rate reached only approximately 30%. The results show that a significant difference (70% of the cumulative drug release rate) was observed between with and without the magnetic stimulations. Results show that each of magnetic stimulations promoted approximately 11–14% of drug release rate in the cumulative drug release. 

Dandamudi et al. revealed that magnetic drug targeting improved localized drug delivery to interstitial tumor targets and tumor vessels. Under magnetic stimulation, the cumulative release rate of drugs in a magnetically responsive control system was faster [17]. Furthermore, according to previous studies, after exposure to a magnetic field, SPIO NPs would oscillate within the biological matrix, and subsequently, compressive and tensile forces would be created to push more loaded drug out of the drug carrier as a pump [47]. In this study, results showed that the timing and numbers of magnetic stimulation could be controlled and designed according to necessity. Therefore, we could control the release timing and dosage of VBL accurately.

## 4. Conclusions

We proposed an in situ approach for manufacturing VBL-loaded SPIO NPs–chitosan composite particles by using a microfluidic chip. This approach could prepare diameter-controllable and uniform sized polymer particles. The fabricated composite particles have a fine spherical shape with diameters ranging from 360 to 420 µm with extremely low R.S.D. (< 7%) in the particle stage. The MTT assay indicated that the composite particles are potential biocompatible materials. The SPIO NPs were employed to perform a magnetically sensitive function of a controlled VBL drug release system. Furthermore, a faster and controllable drug release behavior could be achieved in a pulsatile external magnetic field. Results showed that it provides approximately 11–14% enhancement of drug release in each magnetic stimulation for 10 min. Results presume that the composite particles could be an actively targeted magnetic drug delivery system which can lead drug carriers to reach the intended site and control the drug release rate, timing, and amount accurately in an oscillating or alternating magnetic field. Based on the droplet microfluidic, we have successful synthesis of monodispersed stimuli-responsive magnetic drug carriers. We presume that the prepared magnetically responsive controlled drug release system can be effectively used in many relevant biological and pharmaceutical applications in the future.

## Figures and Tables

**Figure 1 pharmaceutics-11-00212-f001:**
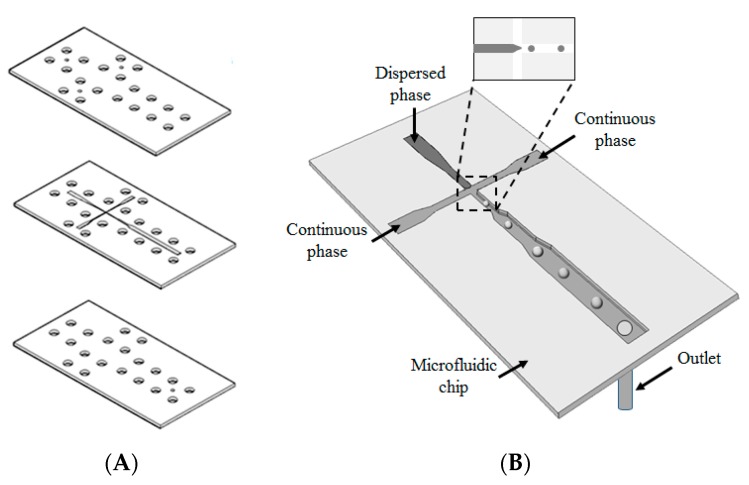
(**A**) Microfluidic chip under an expanded view, and (**B**) schematic drawing of microchannel design. The cross-junction microchannel design could exert a focusing force if two immiscible streams converged, and uniform self-assembling spheres could be obtained subsequently. The broadened channel design in the downstream is used to reduce the flow rate of fluids, and then it can enhance the real-time observation of droplet formation and transportation by a microscope.

**Figure 2 pharmaceutics-11-00212-f002:**
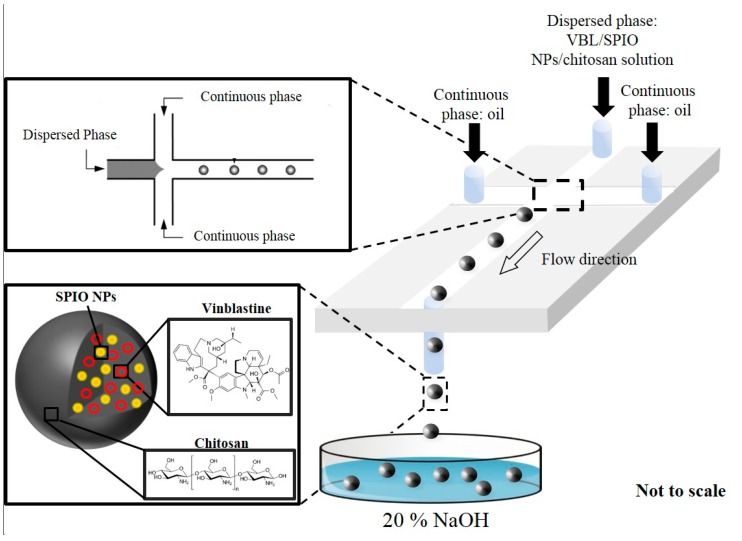
Schematic diagram of the composite polymer particles preparation by a microfluidic channel device. The VBL-loaded SPIO NPs–chitosan emulsions were typically generated in a cross-junction microchannel; the emulsions/droplets were subsequently transported to a 20% NaOH solution through the microchannel. After solidification, VBL-loaded SPIO NPs–chitosan particles were obtained.

**Figure 3 pharmaceutics-11-00212-f003:**
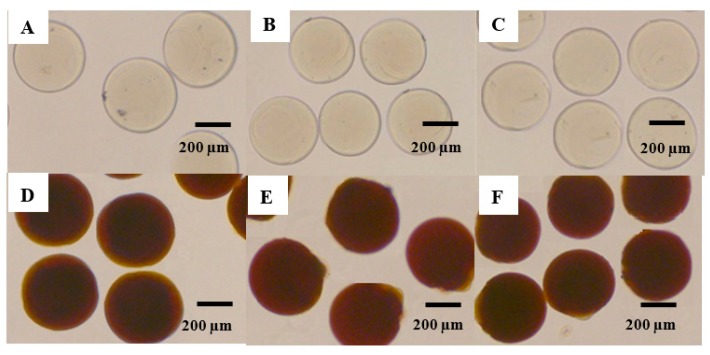
Optical microscopy images of the chitosan composite particles. The (**A**–**C**) images show the VBL-loaded chitosan particles. The (**D**–**F**) images show the VBL-loaded SPIO NPs–chitosan composite particles.

**Figure 4 pharmaceutics-11-00212-f004:**
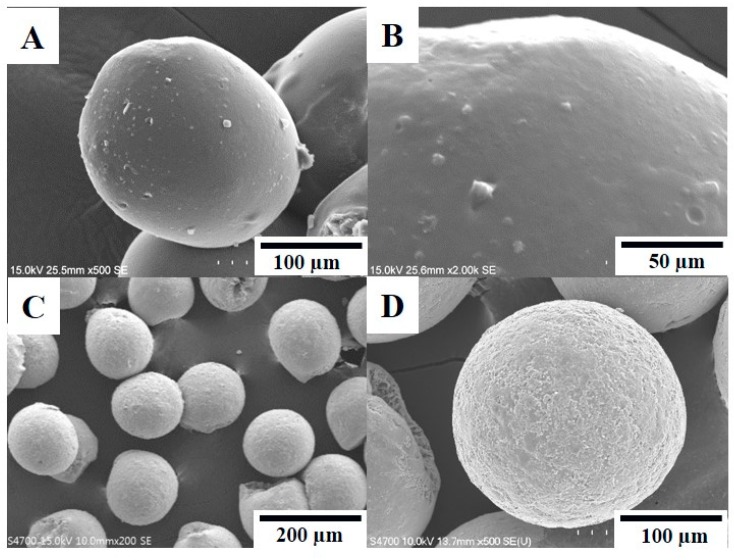
The (**A**,**B**) are SEM images of VBL-loaded chitosan composite particles. The (**C**,**D**) are SEM images of VBL-loaded SPIO NPs–chitosan composite particles.

**Figure 5 pharmaceutics-11-00212-f005:**
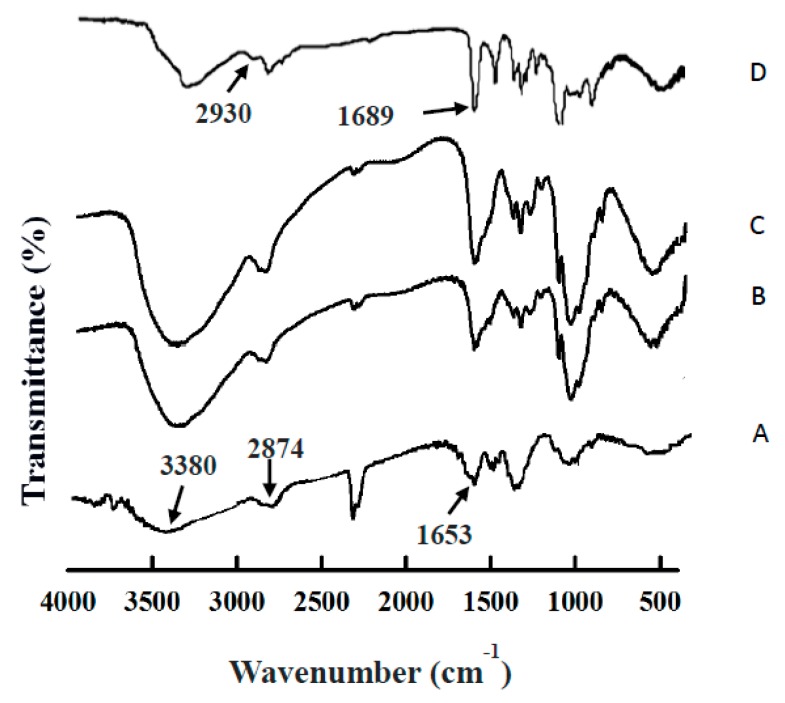
FTIR spectra of the (**A**) chitosan particles, (**B**) SPIO NPs–chitosan composite particles (chitosan MW, 150,000), (**C**) SPIO NPs–chitosan composite particles (chitosan MW, 40,000), and (**D**) VBL-loaded SPIO NPs–chitosan composite particles (chitosan MW, 150,000).

**Figure 6 pharmaceutics-11-00212-f006:**
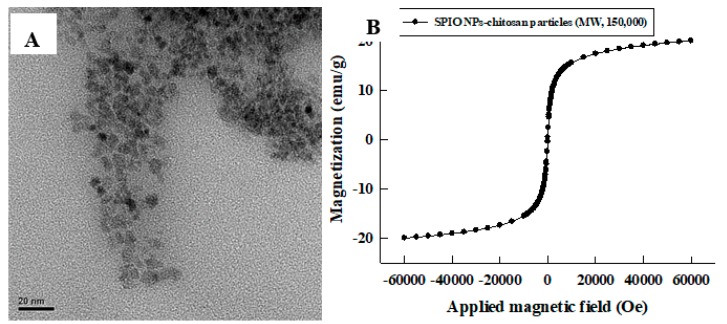
The (**A**) TEM image (scale bar is 20 nm) and (**B**) SQUID data of the synthesized SPIO NPs–chitosan particles.

**Figure 7 pharmaceutics-11-00212-f007:**
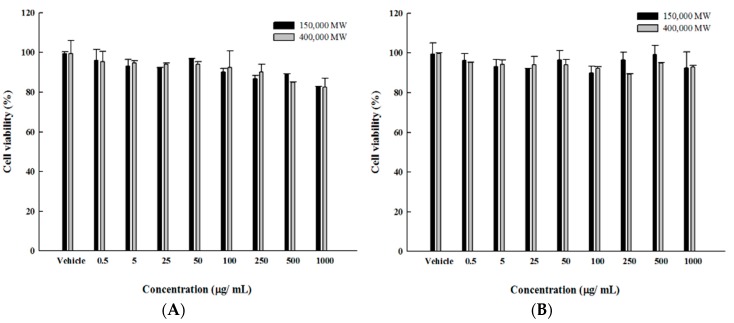
The cell viability of the SPIO NPs–chitosan composite particles in (**A**) 3T3 Cell line and (**B**) MCF-7 cell line.

**Figure 8 pharmaceutics-11-00212-f008:**
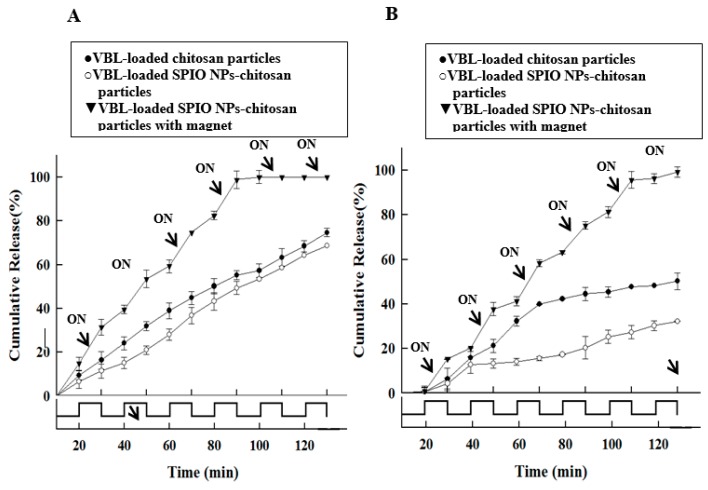
The effect of polymer molecular weight (MW) on drug release patterns, and with or without pulsatile magnetically triggered tests of VBL-loaded SPIO NPs–chitosan composite particles in PBS (pH 7.2). The (**A**) is using 150,000 MW chitosan and (**B**) is using 400,000 MW chitosan.

**Table 1 pharmaceutics-11-00212-t001:** The flow rates influence to the diameter of emulsions and particles.

Flow Rate of Continuous Phase (mL/min)	Flow Rate of Dispersed Phase (mL/min)	Emulsions			Particles			Shrinking (%)
		Average Size (µm)	S.D. ^a^ (µm)	R.S.D. ^b^ (%)	Average Size (µm)	S.D. ^a^ (µm)	R.S.D. ^b^ (%)	
0.8	0.008	550.3	10.8	1.6	390.3	24.7	6.2	27.4
	0.006	544.9	8.6	1.3	385	20.8	5.3	28.9
	0.004	531.7	11.6	1.7	372.5	20.1	5.6	29.9
	0.002	504.2	11.1	1.7	359.4	12.8	3.6	28.7
0.7	0.008	553.5	6	0.9	397	17.5	4.4	28.3
	0.006	546.2	5.4	0.8	389.3	17	4.4	30.4
	0.004	534	10.4	1.5	376.1	13.8	3.7	29.6
	0.002	508	10.2	1.6	375	17.2	4.6	25.7
0.6	0.008	574	10.8	1.4	408.6	18.9	4.7	29.5
	0.006	561.6	3.1	0.4	403.5	21	5.2	28.2
	0.004	552	7.7	1.1	398.8	19.8	5	27.8
	0.002	547.8	13.8	2	389.3	22.1	5.7	28.9
0.5	0.008	585	17.9	2.4	423.1	11.9	2.8	27.7
	0.006	578.5	20.7	2.8	421.6	14.3	3.4	27.1
	0.004	560.6	27.1	3.8	416.9	13.1	3.2	25.6
	0.002	554.3	21.5	3.1	406.4	20.9	5.1	26.7

Note a: standard deviation (S.D.). Note b: relative standard deviation (R.S.D.).

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
