# Peer review of "Microfluidic Synthesis of Vinblastine-Loaded Multifunctional Particles for Magnetically Responsive Controlled Drug Release"

_pharmaceutics, 2019, doi:10.3390/pharmaceutics11050212_

Round 1

Reviewer 1 Report

This paper reports fabrication of a novel microfluidic devide to produce vinblastin-loaded microparticles. The paper is well-written, the results shown are interesting and important. The paper should be revised, however:

- the literature review does not show the full range of applications of relevant magntic nanoparticles-based technologies. The following papers are suggested for citation:

Chen and Liu, https://doi.org/10.1246/bcsj.20180101

German et al, https://doi.org/10.1016/j.colsurfb.2015.07.042 

Zeleňák et al, http://dx.doi.org/10.1039/C8RA05576A 

- all figures are of very low quality, please provide better resolution images. 

- the authors claim that the capsules prepared are biocompatible. This claim is not fully supproted, just a single proliferation test was used. the authors should downplay this claim

- drug delivery is also claimed, though there is no demonstration of any cellular uptake.

Author Response

Comments and Suggestions for Authors

This paper reports fabrication of a novel microfluidic devide to produce vinblastin-loaded microparticles. The paper is well-written, the results shown are interesting and important. The paper should be revised, however:

Response:

Thank you very much for your valuable recommendations.

Question 1

- the literature review does not show the full range of applications of relevant magntic nanoparticles-based technologies. The following papers are suggested for citation:

Chen and Liu, https://doi.org/10.1246/bcsj.20180101

German et al, https://doi.org/10.1016/j.colsurfb.2015.07.042 

Zeleňák et al, http://dx.doi.org/10.1039/C8RA05576A 

Response:

Thank you very much for your valuable recommendations. We have added three references in the revised manuscript.

Revision:

Please see Para #61, refs. 20-22.

20.  Chen, H.; Liu, L., One-step synthesis of polyethylenimine-coated Fe3O4 superparamagnetic nanoparticles for latent fingermark enhancement. B Chem Soc Jpn 2018, 91 (8), 1319-24.

21.  German, S. V.; Navolokin, N. A.; Kuznetsova, N. R., Zuev, V. V.; Inozemtseva, O. A.; Anis'kov, A. A.; Volkova, E. K.; Bucharskaya, A. B.; Maslyakova, G. N.; Fakhrullin, R. F.; Terentyuk, G. S.; Vodovozova, E. L.; Gorin, D. A., Liposomes loaded with hydrophilic magnetite nanoparticles: Preparation and application as contrast agents for magnetic resonance imaging. Colloids Surf B Biointerfaces 2015, 135 (1), 109-15.

22.  Zeleňák, V.; Zeleňáková, A.; Kapusta, O.; Hrubovčák, P.; Girman, V.; Bednarčík, J., Fe2O3 and Gd2O3 nanoparticles loaded in mesoporous silica: insights into influence of NPs concentration and silica dimensionality. RSC Adv 2019, 9, 3679-87.

Question 2

- all figures are of very low quality, please provide better resolution images. 

Response:

Thank you very much for your valuable recommendations. We have revised two figures with high quality in the revised manuscript.

Revision:

Please see revised Figures 1 and 2.

A                      B         

Figure 1. (A) Microfluidic chip under an expanded view, and (B) schematic drawing of microchannel design. The cross-junction microchannel design could exert a focusing force if two immiscible streams converged, and uniform self-assembling spheres could be obtained subsequently. The broadened channel design in the downstream is used to reduce the flow rate of fluids, and then it can enhance the real-time observation of droplet formation and transportation by a microscope.

Figure 2. Schematic diagram of the composite polymer particles preparation by a microfluidic channel device. The VBL-loaded SPIO NPs–chitosan emulsions were typically generated in a cross-junction microchannel; the emulsions/droplets were subsequently transported to a 20% NaOH solution through the microchannel. After solidification, VBL-loaded SPIO NPs–chitosan particles were obtained.

Question 3

- the authors claim that the capsules prepared are biocompatible. This claim is not fully supproted, just a single proliferation test was used. the authors should downplay this claim

Response:

Thank you very much for your valuable recommendations. We have revised “biocompatible” to be “potential biocompatible materials” in the revised manuscript.

Revision:

Please see Para #273, #275, and #325.

Question 4

- drug delivery is also claimed, though there is no demonstration of any cellular uptake.

Response:

Thank you very much for your valuable recommendations. Generally, the microfluidics chip technique is expert in micrometer scale droplets manufacture. In this study, the micrometer scale chitosan particles could be potential used in oral administration. Capsules in micro-scale are too big for cells in the cellular uptake test. In order to overcome the restrictions, in the future we will try to employ nanofluidics chip technique for nano-scale capsules manufacture.

Reviewer 2 Report

The manuscript entitled Microfluidic synthesis of vinblastine-loaded multifunctional particles for magnetically responsive controlled drug release reported a magnetically responsive controlled drug release system designed by embedding superparamagnetic iron oxide (SPIO) nanoparticles (NPs) in a chitosan matrix and an external magnet. The infrared characterization, drug encapsulation efficiency, biosafety of composite nanoparticles, the cytotoxicity and the controlled release application of magnetically responsive drugs have been investigated in a systematic manner. The data is convincible. And this manuscript could be considered acceptance in a major revision after resolving the problems below:

1. The control group of the chitosan particles should be given to demonstrate whether the roughness of the particle surface is different from VBL-loaded chitosan particles and VBL-loaded SPIO NPs-chitosan composite particles.

2. In order to further observe the changes in the internal structure of chitosan particles after loading VBL and Fe3O4 NPs, the TEM images of chitosan particles, VBL-loaded chitosan particles and VBL-loaded SPIO NPs-chitosan composite particles should be confirmed.

3.Could the authors provide the drug encapsulation efficiency?

4. The legend of C and D is missing in the Figure 4. Compared with the Figure 3 and Table 1, the size of microparticle in Figure 4 is obviously smaller.

Author Response

Review #2

English language and style

( ) Extensive editing of English language and style required 
( ) Moderate English changes required 
(x) English language and style are fine/minor spell check required 
( ) I don't feel qualified to judge about the English language and style 

Yes

Can be   improved

Must be   improved

Not applicable

Does the   introduction provide sufficient background and include all relevant   references?

(x)

( )

( )

( )

Is the   research design appropriate?

(x)

( )

( )

( )

Are the   methods adequately described?

(x)

( )

( )

( )

Are the   results clearly presented?

( )

(x)

( )

( )

Are the   conclusions supported by the results?

( )

(x)

( )

( )

Comments and Suggestions for Authors

The manuscript entitled Microfluidic synthesis of vinblastine-loaded multifunctional particles for magnetically responsive controlled drug release reported a magnetically responsive controlled drug release system designed by embedding superparamagnetic iron oxide (SPIO) nanoparticles (NPs) in a chitosan matrix and an external magnet. The infrared characterization, drug encapsulation efficiency, biosafety of composite nanoparticles, the cytotoxicity and the controlled release application of magnetically responsive drugs have been investigated in a systematic manner. The data is convincible. And this manuscript could be considered acceptance in a major revision after resolving the problems below:

Response:

Thank you very much for your valuable recommendations.

Question 1

-The control group of the chitosan particles should be given to demonstrate whether the roughness of the particle surface is different from VBL-loaded chitosan particles and VBL-loaded SPIO NPs-chitosan composite particles.

Response:

Thank you very much for your valuable recommendations. We would like to show the SEM image of chitosan, however the request for SEM image usually take above 3 weeks in our school. This is a long line for the request for SEM image. The defense time of this paper is within 10 days. Therefore, we try to search literature for the SEM image of chitosan. We find that the SEM image of chitosan was published before by our team. Figure 2b (left) show the SEM image of chitosan [45]. Based on the Figure 2b (left) [45], Fig. 4A, and Fig. 4D, results show that chitosan, VBL-loaded chitosan composite particles, and VBL-loaded chitosan composite particles with SPIO NPs were surface smoothness.

Figure 2. Morphologies of the synthesyzed chitosan spheres. (a) An optical image of the iron oxide nanoparticles loaded chitosan spheres. (b) SEM images of chitosan spheres without (left) and with (right) iron oxide nanoparticles. (c) and (d) show the cross-section images of chitosan spheres (without iron oxide nanoparticles). Inset in (c) shows the sliced hemispheres of chitosan sphere. Inset in (d) shows details of the internal structure. (e) and (f) show the cross-section images of iron oxide nanoparticles loaded chitosan spheres [45].

[45] Chih-Hui Yang*, Chih-Yu Wang, Keng-Shiang Huang, Chen-Sheng Yeh, Andrew H.-J. Wang, Wei-Ting Wang, Ming-Yu Lin, Facile synthesis of radial-like macroporous superparamagnetic chitosan spheres with in-situ co-precipitation and gelation of ferro-gels. PloS One, 2012, 7, e49329.

Question 2

-In order to further observe the changes in the internal structure of chitosan particles after loading VBL and Fe3O4 NPs, the TEM images of chitosan particles, VBL-loaded chitosan particles and VBL-loaded SPIO NPs-chitosan composite particles should be confirmed.

Response:

Thank you very much for your valuable recommendations. We have published the internal structure of chitosan particles after loading Fe3O4 NPs in 2012 [45]. The SEM images were employed for the internal structure observation. In this study, chitosan particles and VBL-loaded chitosan particles were micro-scale in diameter. Generally, the particles in micro-scale size are too big for TEM images observation. In order to overcome the restrictions, in the future we will try to employed nanofluidics chip technique for nano-scale capsules manufacture.

[45] Chih-Hui Yang*, Chih-Yu Wang, Keng-Shiang Huang, Chen-Sheng Yeh, Andrew H.-J. Wang, Wei-Ting Wang, Ming-Yu Lin, Facile synthesis of radial-like macroporous superparamagnetic chitosan spheres with in-situ co-precipitation and gelation of ferro-gels. PloS One, 2012, 7, e49329.

Question 3

-Could the authors provide the drug encapsulation efficiency?

Response:

Thank you very much for your valuable recommendations. Based on the high performance liquid chromatography data, the VBL drug encapsulation efficiency was 67%–69%.

Revision:

Please see Para #21 and #264 in the revised manuscript.

Question 4

The legend of C and D is missing in the Figure 4. Compared with the Figure 3 and Table 1, the size of microparticle in Figure 4 is obviously smaller.

Response:

1)      Thank you very much for your valuable recommendations. We have added “C” and “D” in Fig. 4.

2)      Yes, hydrogel microparticles would be smaller after drying. For SEM image test, all hydrogels should be dried.

Round 2

Reviewer 2 Report

The authors have reasonablely addressed the questions and given a convincing explanation. There is a minor question needed to be answered before publication.

In the section of 3.3. Biocompatibility test and in vitro release study, the measured result of MTT assay should be given, not just described as “the results revealed high viability (> 80%)”.

Author Response

Q: In the section of 3.3. Biocompatibility test and in vitro release study, the measured result of MTT assay should be given, not just described as “the results revealed high viability (> 80%)”.

A: Thank you very much for your valuable recommendations. We have added the cell viability test data in the revised manuscript, please see Fig. 7.